# The Pseudolesions of the Oral Mucosa: Differential Diagnosis and Related Systemic Conditions

**Fedora della Vella** [1,*], **Dorina Lauritano** [2], **Carlo Lajolo** [3], **Alberta Lucchese** [4],
**Dario Di Stasio** [4], **Maria Contaldo** [4], **Rosario Serpico** [4] **and Massimo Petruzzi** [1,*]

1   Interdisciplinary Department of Medicine, University of Bari "Aldo Moro", 70124 Bari, Italy
2   School of Medicine and Surgery, University of Milano-Bicocca, 20900 Monza, Italy;
    dorina.lauritano@unimib.it
3   Department of Head and Neck, Oral Surgery and Implantology Unit, University Cattolica del Sacro Cuore,
    00168 Rome, Italy; carlo.lajolo@unicatt.it
4   Multidisciplinary Department of Medical-Surgical and Dental Specialties, University of Campania "Luigi
    Vanvitelli", 80138 Naples, Italy; alberta.lucchese@unicampania.it (A.L.);
    dario.distasio@unicampania.it (D.D.S.); maria.contaldo@gmail.com (M.C.);
    rosario.serpico@unicampania.it (R.S.)
*   Correspondence: dellavellaf@gmail.com (F.d.V.); massimo.petruzzi@uniba.it (M.P.);
    Tel.: +39-0805478388 (F.d.V.)

**Abstract:** Pseudolesions are defined as physiological or paraphysiological changes of the oral normal anatomy that can easily be misdiagnosed for pathological conditions such as potentially malignant lesions, infective and immune diseases, or neoplasms. Pseudolesions do not require treatment and a surgical or pharmacological approach can constitute an overtreatment indeed. This review aims to describe the most common pseudolesions of oral soft tissues, their possible differential diagnosis and eventual related systemic diseases or syndromes. The pseudolesions frequently observed in clinical practice and reported in literature include Fordyce granules, leukoedema, geographic tongue, fissured tongue, sublingual varices, lingual fimbriae, vallate papillae, white and black hairy tongue, Steno's duct hypertrophy, lingual tonsil, white sponge nevus, racial gingival pigmentation, lingual thyroid, and eruptive cyst. They could be misdiagnosed as oral potential malignant disorders, candidiasis, Human Papilloma Virus (HPV)-related affections, oral autoimmune diseases, or benign and malignant tumors. In some cases, pseudolesions feature in a syndromic panel, for example, fissured tongue in Melkersson–Rosenthal syndrome. It is strictly fundamental for dentists to know and to distinguish oral pseudolesions from pathological conditions, in order to avoid overtreatment.

**Keywords:** oral pseudolesion; geographic tongue; Fordyce granules; differential diagnosis

## 1. Introduction

Oral lesions are characterized by tissue alterations, associated with cytological and histological changes [1]. They can be determined by traumatic, infective, immune, potentially neoplastic, and neoplastic (benign or malignant) processes that affect the oral mucosa with different clinical appearance, onset time, and intensity. Generally, dentists point out oral lesions noting changes in size, surface morphology, and/or color of an oral mucosal area compared to the surrounding healthy mucosa [2].

Pseudolesions are instead normal oral anatomical structures or paraphysiologic changes of the oral mucosa with no pathological significance that on routine oral examination may be misdiagnosed as pathological alterations [3].

Pseudolesions can trick not only the patient, causing apprehension and cancer phobia, but also the clinician. A surgical or medical approach to these conditions is not only useless but can result in overtreatment procedures.

Some of these pseudolesions can simulate oral potentially malignant lesions, vascular abnormalities, infective and autoimmune diseases, or neoplasms. Their identification is important in order to establish a correct differential diagnosis from oral diseases and to avoid inappropriate medical or surgical treatments. In fact, pseudolesions do not require any treatment or therapy [4].

The aim of this review is to illustrate the most common oral pseudolesions, their possible differential diagnosis, and their eventual association with syndromes and systemic diseases.

## 2. Fordyce Granules

Small yellow "dust-like" bilateral granules disseminated on the oral mucosa are usually observed in about 80% of individuals. They are ectopic sebaceous glands with no pathological significance [5]. The lips edges, the vestibular mucosa, and the retromolar area are the most involved oral sites (Figure 1A). Fordyce granules do not give any symptoms, except for a rough mucous sensation. They are often mistaken for a fungal infection or lichen planus papules. No treatment is needed.

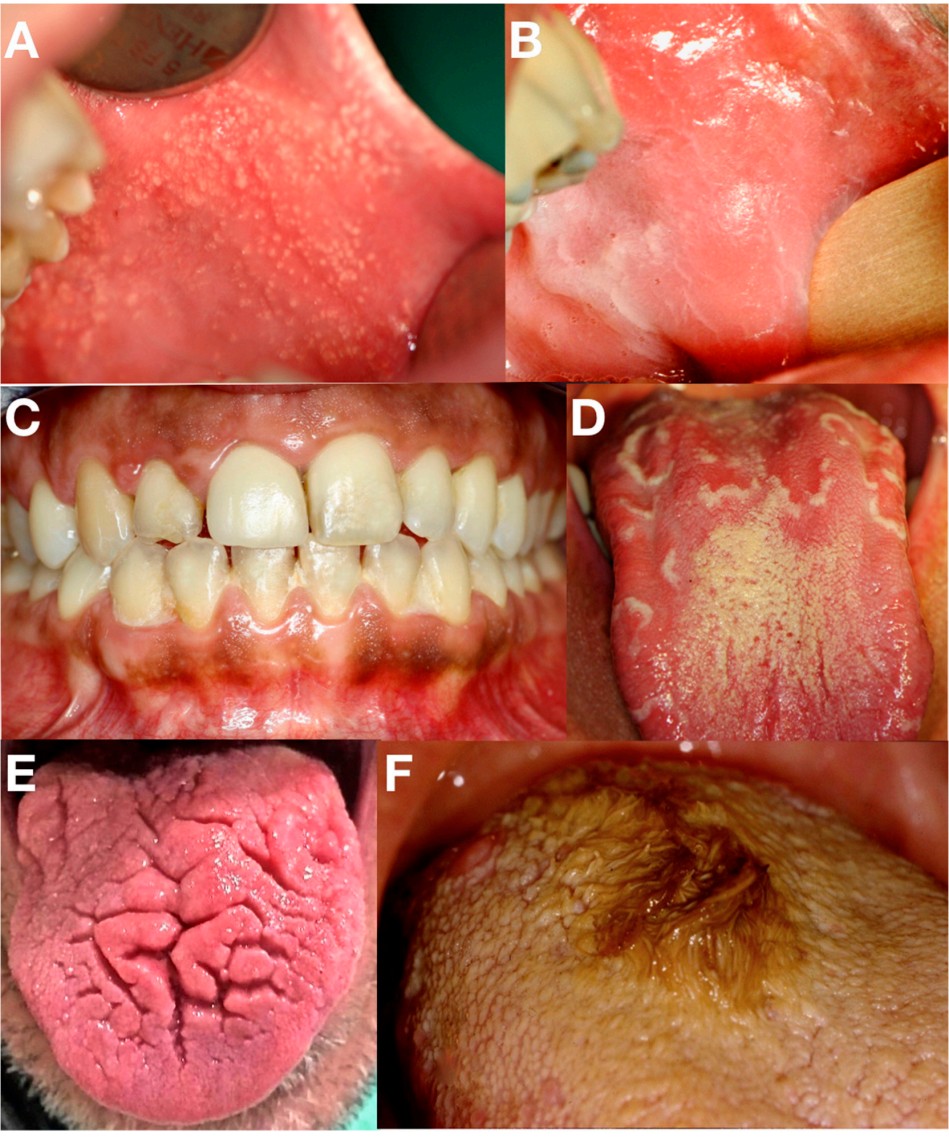

**Figure 1.** (**A**) Fordyce granules; (**B**) leukoedema; (**C**) gingival pigmentation; (**D**) geographic tongue; (**E**) fissured tongue; (**F**) black hairy tongue.

Interestingly, an association between Lynch syndrome (non-polyposic colorectal carcinoma syndrome) and the Fordyce granules seems to exist, explainable by the activation of a pathway responsible both for the development of neoplasia and the activation of the sebaceous glands [6].

Careful observation of Fordyce granules, especially located in lower gingival and vestibular mucosa, may be feasible to identify families potentially affected by non-polyposic colorectal carcinoma syndrome.

## 3. Leukoedema

It is characterized by an opalescent white appearance of the vestibular and buccal mucosa (Figure 1B). The causes are unknown. It is very common in black people (90% of individuals) while rarely found in Caucasians. It is not considered a lesion, but a variation of the normal anatomy of the oral mucosa due to intra- and extracellular imbibition [7]. Leukoedema is always bilateral, and it characteristically disappears when the cheek is stretched only to reappear after releasing the mucosa (diascopic phenomenon); this makes it well distinguishable from leukoplakia or *morsicatio buccarum*. Leukoedema does not require any treatment [8].

## 4. White Sponge Nevus

It is a rare genodermatosis affecting 1:200,000 people [9], transmitted by an autosomal dominant character with high penetrance. It has been shown to be related to keratin defects, because of mutations in the genes encoding mucosal-specific keratins K4 and K13. It looks like a white plaque with velvety or villous appearance, localized on the vestibular mucosa of both sides. The lesion often extends to the tongue, the mouth floor, and the oropharynx mucosa, in some cases genital mucosae are also involved [10]. It can resemble a proliferative leukoplakia or a hyperkeratosic oral lichen planus. It may need surgical removal in case of oral discomfort [11]. There is only one case reporting an association between white sponge nevus and ectrodactyly–ectodermal dysplasia–clefting (EEC) syndrome [12].

## 5. Physiologic (Ethnic/Racial) Gingival Pigmentation

Physiologic pigmentation develops during the first two decades of life. Pigmentation is asymptomatic and no treatment is required except for aesthetic concerns. Moreover, color variation may be uniform, unilateral, bilateral, mottled, macular, or blotched and may either involve the gingival papillae alone, or extend throughout the gingiva and into other oral tissues [13]. Physiologic pigmentation clinically appears as multifocal or diffuse, with variable prevalence in different ethnic groups. It is common in African, Asian, and Mediterranean populations, and it is due to an increased melanocyte activity rather than due to a greater number of melanocytes. Attached gingiva is the most common site of such pigmentation (Figure 1C) [14].

It can easily be misdiagnosed as melanosis, smoking pigmentation, or melanoma, which generally appear as localized mucosal stains; although, in the case of pigmentation due to trauma and smoking, the anamnestic data and the oral sites involved, can be suggestive of their etiology. Gingival and oral pigmentations are also typical of some systemic syndromes, such as Addison's disease, Peutz–Jeghers, McCune–Albright, and Laugier–Hunziker disease [15–17].

## 6. Geographic Tongue (Migrant Glossitis, Migrant Erythema)

It is a benign condition usually observed on the dorsal tongue mucosa (rarely on other oral mucosal sites). It is reported in 1–3% of the healthy population with no difference between females and males. All ages are affected. The etiopathogenesis remains completely unknown. Hypersensitivity to foods or to other substances has been hypothesized but never definitively demonstrated. Diabetes, psoriasis, and hormonal changes are described as possibly associated with this condition.

The turnover mechanism that regulates physiological tongue desquamation is probably impaired, causing the persistence of hypermature areas (the white ones) concurrent with atrophic and hypomature areas (the red ones).

A typical whitish border surrounds the erythematous area (Figure 1D). The lesions spontaneously regress and then reappear after days or weeks. In several cases, geographic tongue is asymptomatic while in some patients it causes burning and discomfort linked to acid, hot, or spicy food ingestion. It can be a source of cancer phobia. It is important to reassure the patient about the absolute benignity of the lesion [18].

This pseudolesion aspect can be misdiagnosed as erythematous candidiasis, erythema multiforme, atrophic erosive lichen planus, and vesiculosus-bullous diseases. Psoriasis, allergy and atopy, diabetes, hypertension, tobacco use, and psychological factors are reported to be associated with geographic tongue [19]. In case of burning sensation and/or soreness affecting the patients' quality of life, topical steroids can be prescribed.

### 7. Fissured Tongue (Scrotal Tongue)

The fissured tongue is a very common condition, affecting about 2–5% of the population, especially adults. It probably has a hereditary background, and it can develop at any age.

The tongue dorsal surface and the margins show depth fissures of variable dimension and depth [20]. Some patients show one deep central fissure only, in other cases, numerous radial fissures, similar to the cerebral sulci or scrotum or walnut husks, radiate from the tongue surface (Figure 1E). The fissured tongue is often associated with the geographic tongue; in this case, superficial areas of erythema surrounded by whitish borders are accompanied by the fissures. The fissured tongue is generally asymptomatic, although in some patients it may cause burning or pain due to spicy and/or acid food intake [21]. Fissures facilitate food stagnation and the proliferation of bacterial and mycotic flora. Together with lip numbness and facial paralysis, fissured tongue is part of the triad of Melkersson–Rosenthal syndrome (orofacial granulomatosis), and it frequently occurs in patients affected by Down syndrome [22].

### 8. White and Black Hairy Tongue

White hairy tongue is characterized by a marked hypertrophy of the filiform papillae on the dorsal surface of the tongue. Pathogenesis is probably linked to excessive keratin production by the tongue epithelium. Other possible etiological factors include excessive smoking, poor oral hygiene, and dysbiosis of oral microflora [23]. This condition is often observed after a prolonged antibiotic therapy. A thick layer of keratin and filiform papillae covers the tongue, giving it a whitish and dried aspect. If present, the black color is due to the production of bacterial pigments or, in smokers, the deposition of nicotinic derivatives (Figure 1F). Some patients complain of a tickling sensation to the soft palate or apprehension due to its unpleasant appearance. It may predispose mycotic infections. The pseudolesion's appearance can be mistaken for candidiasis. It is useful to recommend smoking suspension, accurate oral hygiene, and lingual brushing to facilitate debris and keratin excess removal [24].

### 9. Hyperplasia of Lingual Fimbriae

Lingual fimbriae are normal anatomical structures that appear as small filiform flanges on the ventral surface of the tongue at the sides of the frenulum (Figure 2A). If hyperplastic, lingual fimbriae can easily induce diagnostic errors, in fact they are usually confused with squamous papillomas or warts [17].

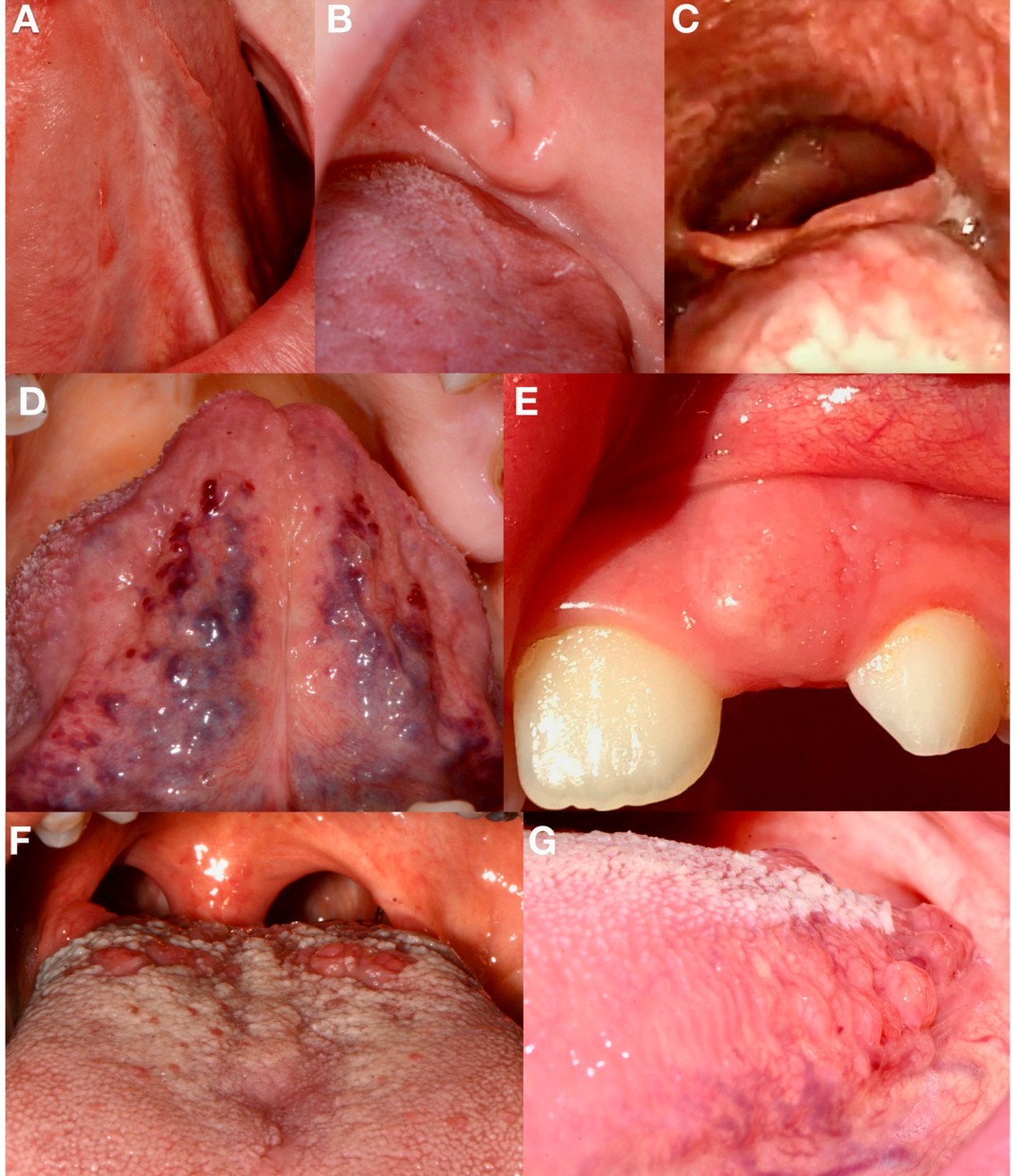

**Figure 2.** (**A**) Lingual fimbriae; (**B**) Steno's duct orifice hyperplasia; (**C**) lingual thyroid (endoscopic picture); (**D**) sublingual varices; (**E**) eruptive cyst; (**F**) vallate papillae; (**G**) lingual tonsil.

## 10. Steno's Duct Orifice Hyperplasia

The parotid gland duct, also called Steno's duct, ends in the oral cavity on the buccal mucosa, facing the vestibular surface of the first or second upper molar. A hyperplasia of the Steno orifice for traumatic or infective causes can occur. It becomes enlarged, assuming the appearance of a minute fibroma (Figure 2B). Failure to recognize the glandular anatomical structure could cause useless biopsy with damage to the glandular structures [25].

## 11. Lingual Thyroid

It is due to the persistence of ectopic thyroid tissue in the posterior portion of the lingual surface (Figure 2C). It is an embryogenetic anomaly caused by a defective migration of the thyroid gland from the primitive pharyngeal cavity to its normal anatomical position [26]. If asymptomatic, the lingual

thyroid does not require any treatment; when necessary, therapy is based on the administration of thyroid hormones for suppressive purposes. Surgical ablation or radioactive iodine is reserved for cases that are not responsive to medical therapy [27]. Rare cases of association between hyperthyroidism and lingual thyroid are reported [28]. Dermoid cyst, tongue neoplasms, and lymphatic malformations can be considered in differential diagnosis; radionuclide scanning is necessary to confirm the presence of ectopic thyroidal tissue [29].

## 12. Sublingual Varices

Sublingual varices are clinically evident as small enlarged veins on the anterior ventral surface of the tongue (Figure 2D). They are related to the aging process and hence to collagen elastic fiber degeneration and weakening of the venous wall. In fact, the prevalence of sublingual varices increases with age, reported in up to 60% of elderly patients, in both sexes, and in different population groups. They might be mistaken for Osler–Weber–Rendu syndrome or hereditary hemorrhagic telangiectasia or multiple hemangiomas. Hypertension, cardiovascular diseases, smoking, and dental wearing are described as associated with sublingual varices [30].

## 13. Eruption Cysts

Eruption cysts are benign, odontogenic developmental cysts associated with a primary or, more often, permanent tooth in the eruptive phase. Clinically, the cyst appears as a soft bluish gingival mass on the alveolar ridge overlying the crown of an erupting tooth (Figure 2E). They are usually asymptomatic, and the related teeth erupt without any complications in approximately two months. The incisive and molar areas are frequently involved [31]. Its aspect can simulate a hemangioma.

Nomura et al. reported multiple eruption cysts in a four-year-old boy with Menkes kinky hair disease in treatment with an anticonvulsant [32].

## 14. Vallate Papillae

Vallate papillae are physiological anatomical structures located on the back of the tongue surface along the sulcus terminalis (Figure 2F). They are usually 8 or up to 12, round with a small central pitting where the lingual Von Ebner's salivary gland ducts end. All papillae protrude about 2 mm above the lingual mucosa, but in some individuals, they can appear more prominent and pronounced; this can be due to subjective constitution, responsive hypertrophy to irritative triggers (such as gastroesophageal reflux), or a generalized atrophy of the dorsal tongue [33–35].

## 15. Lingual Tonsils

Lingual tonsils are collections of lymphoid tissue placed at the back of the tongue, one on either side, often associated with foliate papillae (Figure 2G). They take part in the formation of Waldeyer's ring. They are less prone to be infected compared to other pharyngeal tonsils, thanks to the presence of mucous tongue glands secreting into the crypts. These structures can become especially visible in case of hypertrophy, that can often occur in mouth breathers and in patients affected by gastroesophageal reflux [36,37]. Due to their localization, they are frequently misdiagnosed as oral carcinoma or benign neoplasms.

## 16. Conclusions

The embryogenetic development of the oral and facial tissues is quite complex, and, in most cases, it is responsible for simple variations in normal healthy oral anatomy.

These variations could be mistaken for pathologies and be a cause of concern for individuals referring to their dentists. Dentists should reassure their patient about the absolute benignity of the observed pseudolesions avoiding any surgical diagnostic procedure.

**Author Contributions:** Conceptualization, F.d.V. and M.P., investigation, F.d.V and M.P., resources, R.S., D.D.S and A.L., data curation, F.d.V. and M.P., validation, M.C., C.L. and D.L., writing-original draft preparation, F.d.V and M.P, writing-review and editing, A.L., D.L., D.D.S. and M.C., visualization, C.L. and D.L., supervision, R.S., D.L. and M.P, project administration M.P., R.S. and C.L.

**Funding:** The research received no external funding.

**Acknowledgments:** The authors are grateful to Professor Nicola Antonio Adolfo Quaranta for providing the lingual thyroid picture and to Niccolò Petruzzi for the eruptive cyst picture.

**Conflicts of Interest:** The authors declare no conflict of interests.

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
