# Peer review of "The Pseudolesions of the Oral Mucosa: Differential Diagnosis and Related Systemic Conditions"

_applsci, doi:10.3390/app9122412_

Round 1
Reviewer 1 Report
Manuscript ID: applsci-511432
Type of manuscript: Review
Title: The pseudolesions of the oral mucosa: differential diagnosis and
related systemic conditions.
This is an interesting review on this topic and it is informative for a dentist. It is well designed and prepared and the figures are in good quality. However, it does not add any new information in literature.
Author Response
30th May 2019
MANUSCRIPT: applsci-511432
Dear Editor,
We would like to thank you and the reviewers for the thoughtful review of our manuscript " The pseudolesions of the oral mucosa: differential diagnosis and related systemic conditions" and the opportunity to revise it. As requested, below we provide a point-by-point response to all comments and questions. We are submitting a corrected version with revisions in blue underlined. We believe these revisions have substantially improved the manuscript and look forward to your decision. We will be happy to make additional revisions if required.
Sincerely,
Dr. Fedora della Vella for all the authors.
Reviewer 1
We thank the reviewer for his comment. As he noted, this is a narrative review with an informative aim for dentists’ daily practice.

Reviewer 2 Report
This paper reviewed several pseudolesions, which are well-known physiological condition. Entirely, it is not strikingly novel but includes some interesting viewpoints which might give some clinical suggestions to readers. Therefore, I think this manuscript can be accepted for publication after minor revision, if this can be suitable for the aims and scope of this journal.
Queries and recommendations
On line 60: I recommend that authors insert a following sentence to realize clinical importance of Fordyce granules abnormally distributing in HNPCC families. “Careful observation of FGs, especially located in lower gingival and vestibular mucosa, may be feasible to identify HNPCC affected families.
Can you add a macroscopic picture of white sponge nevus?
As you showed on line 88, physiological pigmentation can be easily misdiagnosed as some pathological conditions, please add brief viewpoints for differential diagnosis.
Fig. 2C looks like to be inadequate focus so that authors should improve this photograph.
Author Response
R:On line 60: I recommend that authors insert a following sentence to realize clinical importance of Fordyce granules abnormally distributing in HNPCC families. “Careful observation of FGs, especially located in lower gingival and vestibular mucosa, may be feasible to identify HNPCC affected families.”
A: Sentence added as suggested.
R:Can you add a macroscopic picture of white sponge nevus?
A: Unfortunately, we do not own pictures of white sponge naevus cases. If the reviewer considers the picture indispensable, we will ask to other journals for permission to use one of their copyrighted published images.
R: As you showed on line 88, physiological pigmentation can be easily misdiagnosed as some pathological conditions, please add brief viewpoints for differential diagnosis.
A: Done as suggested.
R:Fig. 2C looks like to be inadequate focus so that authors should improve this photograph
A:This is an endoscopic picture kindly provided by a colleague, as we express in the acknowledgment. We changed the figure’s caption specifying the endoscopic acquisition method. If the quality is considered unacceptable, we will remove the lingual thyroid picture.
